**Original Research Article**

drought stress; meta-analysis; *Oryza sativa*; RNA-sequencing; Salt stress.

**Corresponding author:**
Hidemasa Bono;
Email: bonohu@hiroshima-u.ac.jp

**Associate Editor:**
Dr. Eunyoung Chae

# Meta-analysis of public RNA-sequencing data of drought and salt stresses in different phenotypes of resistant and susceptible *Oryza sativa* cultivars

Mitsuo Shintani[1] and Hidemasa Bono[1,2]

[1]Graduate School of Integrated Sciences for Life, Hiroshima University, Higashi-Hiroshima, Japan; [2]Genome Editing Innovation Center, Hiroshima University, Higashi-Hiroshima, Japan

## Abstract

Environmental stresses, such as drought and salt, adversely affect plant growth and crop productivity. While many studies have focused on established components of stress signaling pathways, research on unknown elements remains limited. In this study, we collected RNA sequencing (RNA-Seq) data from *Oryza sativa* registered in public databases and conducted a meta-analysis integrating multiple studies. We analyzed 105 paired RNA-Seq datasets from resistant or susceptible *O. sativa* cultivars under salt and drought conditions to identify novel stress-responsive genes with common expression changes across these datasets. A meta-analysis identified 10 genes specifically upregulated in resistant cultivars and 12 specifically upregulated in susceptible cultivars under both drought and salt conditions. Furthermore, by comparing previously identified stress-responsive genes in *Arabidopsis thaliana*, we explored genes potentially involved in stress response mechanisms that are conserved across plant species. The genes identified in this data-driven study may serve as targets for future research and genome editing.

## 1. Introduction

Rice (*O. sativa*) is an important food in East Asia, South Asia, the Middle East, Latin America, and the West Indies, and it is estimated to account for more than one-fifth of the calories consumed by humans worldwide (Sharif et al., 2014). Rice is a crucial crop because it is high in calories and contains more abundant amounts of essential vitamins and minerals than other grains (Mohidem et al., 2022). However, rice production faces salinity and drought stress, which hinders its cultivation.

Salinity and drought are major abiotic stresses that inhibit plant growth and productivity (Angon et al., 2022; Billah et al., 2021; Kamran et al., 2019; Ma et al., 2020; Yang & Guo, 2018). Rice is susceptible to the negative effects of drought and salt damage, which lead to serious issues of reduced yields. Twenty percent of agricultural land currently used worldwide is affected by salt stress, and this percentage increases every year owing to anthropogenic and natural factors (C. Liu et al., 2022). High Na + levels induce K+ and Ca$_2$+ efflux from the cytoplasm, causing intracellular homeostatic imbalance, nutrient deficiency, oxidative stress, growth retardation, and cell death (Ahanger & Agarwal, 2017). In addition, high salt concentrations reduce photosynthesis through stomatal limitations, such as stomatal closure (Munns & Tester, 2008). Salt stress also has a negative effect on photosynthesis because of non-stomatal limitations, such as chlorophyll dysfunction, deficiency of photosynthetic enzyme proteins and membranes, and disruption of the ultrastructure of the chloroplast (Gengmao et al., 2015; Jiang et al., 2012; Mittal et al., 2012).

Drought represents a state of soil water deficit in which insufficient water is available for plants to fully grow and complete their life cycles. As soil water evaporates, salinity is concentrated in the soil solution, causing drought and salinity simultaneously ("Impacts of soil salinity/sodicity on soil-water relations and plant growth in dry land areas: A review," 2021).

Across Asia, 20% of all rice-producing areas are affected by drought each year (Gowda et al., 2011). Several studies focusing on plant drought have shown that drought delays the flowering time of rice plants and consequently suppresses plant growth, resulting in reduced ear number and fertile fruit number, ultimately reducing yield (Ekanayake et al., 1989; Liu et al., 2010; Yue et al., 2005; Zou et al., 2007). Drought stress causes changes in the antioxidant and osmotic regulatory systems of rice, leading to the accumulation of antioxidants and osmotic regulators (Ouyang et al., 2010; Wang et al., 2019).

Previous studies identified several genes that contribute to salinity and drought stress in rice (Hassan et al., 2023; Liu et al., 2022; Panda et al., 2021). For example, the transcription factors *OsMYBc*, *OsbZIP71*, and *OsNF-YC13* improve salt tolerance by activating K+/Na + transporters and increasing the K+/Na + ratio (Liu et al., 2014; Manimaran et al., 2017; Wang et al., 2015).

In addition, *OsP5CS1* and *OsP5CR* play crucial roles in synthesizing compatible solute proline, increasing proline levels, and thereby enhancing salt tolerance (Sripinyowanich et al., 2013). *OsCPK4* and *OsCPK12*, which encode calcium-dependent protein kinases, are involved in scavenging reactive oxygen species (ROS), thereby enhancing salt tolerance (Asano et al., 2012; Campo et al., 2014). Similarly, the transcription factors *OsZFP179* and *OsZFP182* also improve salt tolerance by enhancing ROS scavenging ability (Huang et al., 2012; Sun et al., 2010).

Regarding drought tolerance, the expression of *OsCPK9* promotes stomatal closure and improves osmotic regulation, thereby enhancing plant drought tolerance (Wei et al., 2017). The transcription factors *OsNAC5* and *OsNAC10* increase grain yield and root development under drought conditions (Jeong et al., 2010, 2013). *OsDREB2A* enhances the survival of transgenic plants under drought stress and *OsDREB2B* increases root length and number (Cui et al., 2011; Matsukura et al., 2010).

However, current research has only revealed a small part of the overall stress response mechanism in rice. In particular, the causes of large differences in stress sensitivity among rice varieties are not fully understood. Therefore, identifying the factors that determine the differences in stress sensitivity is important for elucidating the entire stress response mechanism in rice.

In this study, we focused on the differences in drought and salt stress responses among rice varieties and attempted to identify genes specifically involved in stress sensitivity (susceptible), stress tolerance (resistant), and each phenotype. Through the integrated analysis of a large amount of RNA sequencing (RNA-Seq) data retrieved from public databases, we explored novel responsive genes that have not been reported to be associated with salt or drought stress. By comparing differentially expressed genes identified between cultivars with different susceptibilities to stress, we narrowed down candidate genes involved in the phenotype.

In a previous study, we performed a meta-analysis of stress conditions involving abscisic acid (ABA), salt, and dehydration in *Arabidopsis thaliana* (Shintani et al., 2024). By comparing the stress-responsive genes identified in *O. sativa* from this study with those from the meta-analysis of *A. thaliana*, we identified candidate genes that contribute to stress response mechanisms conserved across plant species.

The findings of this study provide a new perspective for basic research on the stress response mechanisms in rice. Moreover, the identified genes have the potential for use as target candidates for genome editing to develop stress-resistant rice.

## 2. Materials and methods

### 2.1. Manual curation of gene expression data from public databases

RNA-Seq data were collected from the National Center for Biotechnology Information Gene Expression Omnibus (NCBI GEO) (https://www.ncbi.nlm.nih.gov/geo/) (Barrett et al., 2013) and European Bioinformatics Institute BioStudies (EBI BioStudies) (https://www.ebi.ac.uk/biostudies/) (Sarkans et al., 2018). In the NCBI GEO, a comprehensive search was performed using the following query: ("drought"[All Fields] OR "water deficit"[All Fields] OR "salt"[All Fields] OR "NaCl"[All Fields] OR "salinity"[All Fields]) AND "O. sativa"[porgn] AND "Expression profiling by high throughput sequencing"[Filter].

In the EBI BioStudies, projects related to stress treatments were selected by searching with the following parameters: "Keywords: 'RNA-Seq' AND 'O. sativa' AND ('salt' OR 'drought')," "Collection: 'Array Express.'"

Manual curation was performed based on the search results. The manual curation process included collecting metadata from the database, such as stress treatment conditions and sample tissues, along with the raw RNA-Seq data. Based on the descriptions in the papers and database metadata, the phenotype of each cultivar was classified as either "resistant" or "susceptible" to stress. Another criterion was the presence of paired stress-treated and stress-untreated control samples in the same project. This process resulted in the creation of 105 paired datasets from 202 samples from 11 projects.

### 2.2. Gene expression quantification

FASTQ-formatted files for each RNA-Seq run accession number were obtained using the prefetch and fastq-dump commands of the SRA Toolkit (v3.0.0) [https://github.com/ncbi/sra-tools]. Quality control of the raw reads was performed using Trim Galore (v0.6.7) [https://github.com/FelixKrueger/TrimGalore]. Transcript quantification was performed using Salmon (v1.8.0) [https://github.com/COMBINE-lab/salmon] (Patro et al., 2017). We tested two reference transcriptomes from Ensembl Plants (release 56), the *O. sativa Japonica* IRGSP-1.0 (Oryza_sativa.IRGSP-1.0.cdna.all.fa) and the *O. sativa Indica* ASM465v1 (Oryza_sativa.ASM465v1.cdna.all.fa), to evaluate potential mapping biases between subspecies. The *Japonica* reference (IRGSP-1.0) consistently offered higher mapping rates for most *Indica* samples, likely due to more comprehensive and well-curated gene annotations. Therefore, to ensure data consistency and enable accurate gene expression comparisons, we chose to use the *Japonica* reference for quantification in both *Japonica* and *Indica* datasets. This approach enhances comparability and unifies annotations. We also examined the assembly and gene annotation documentation for both references (https://plants.ensembl.org/Oryza_sativa/Info/Annotation/#genebuild, https://plants.ensembl.org/Oryza_indica/Info/Annotation/#genebuild). The *Japonica* reference (IRGSP-1.0) has been continuously updated and curated since its assembly in 2015, whereas the *Indica* reference (ASM465v1) has remained largely unchanged since its assembly in 2011 and lacks recent annotations. These differences in annotation quality may explain the higher mapping rates observed when aligning *Indica* samples to the *Japonica* reference. Consequently, we chose the *Japonica* reference to ensure comprehensive, more recent gene annotations.

As a result, the quantitative RNA-Seq data were calculated as the TPM. To further consolidate the data for downstream analysis, we employed the tximport package (v1.26.1) to aggregate transcript-level TPM values into gene-level expression measurements. Specifically, we generated a one-to-one correspondence table linking transcript IDs (e.g., Os01t0101600–01, −02, −03) with their corresponding gene IDs (e.g., Os01g0101600) and then aggregated the TPM values of multiple isoforms belonging to the same gene by summing them.

### 2.3. TN ratio and score calculation

Gene expression data from different experiments were normalized by calculating the TN ratio, which represents the ratio of gene expression between stress-treated (T) and non-treated (N) samples. If the TN ratio was higher than the threshold, the gene was considered to be upregulated; whereas if it was less than the reciprocal of the threshold, the gene was considered downregulated. Otherwise, the gene was considered unchanged. To classify the upregulated and downregulated genes, 1.5-fold, 2-fold, 5-fold, and 10-fold thresholds were evaluated, and a 2-fold threshold was adopted. Therefore, genes with a TN ratio higher than 2 were classified as upregulated, while genes with a TN ratio lower than 0.5 were classified as downregulated. The TN score of each gene was determined by subtracting the number of downregulated experiments from the number of upregulated experiments to assess changes in gene expression under stress conditions across multiple experiments. The TN ratio and TN score were computed using code that had been utilized in a previous study (Ono & Bono, 2021).

### 2.4. DESeq2 analysis

To complement our TN-score method, we employed DESeq2 (v1.46.0), a robust statistical tool widely recognized for identifying differentially expressed genes from RNA-seq data. Raw read counts obtained from Salmon quantification were used as input for the analysis, which compared salt-treated and untreated samples of salt-resistant cultivars. Significant expression changes were defined as at least twofold upregulation or downregulation (i.e., $\log^2 \text{FoldChange} \geq |1|$). We applied multiple false discovery rate (FDR) thresholds – padj <0.05, 0.01, 0.005, and 0.001 – to identify statistically significant changes. In subsequent analyses, we adopted the FDR threshold of padj <0.05.

### 2.5. Gene set enrichment analysis

Gene set enrichment analysis was performed using the web tool ShinyGO (v0.77) [http://bioinformatics.sdstate.edu/go77/] (Ge et al., 2020) to analyze the DEG sets. For the *O. sativa* analysis, the species was set to "*O. sativa Japonica* Group," and the pathway database was set to "GO Biological Process." All other parameters were set to the default settings.

### 2.6. Visualization

A web-based Venn diagram tool [https://bioinformatics.psb.ugent.be/webtools/Venn/] and the UpSet plot tool [https://asntech.shinyapps.io/intervene/] (Khan & Mathelier, 2017) were used to search for and visualize overlapping genes.

### 2.7. Functional annotation of O. sativa genes

To functionally annotate the *O. sativa* genes, we retrieved the "Gene ID," "Gene name," and "Gene description" for each gene from the Ensembl Plants (release 56, *O. sativa Japonica* Group genes (IRGSP-1.0)). We then identified the putative orthologs of these genes in *A. thaliana* by performing a BLASTP (v2.12.0) search using the *O. sativa* protein sequences (*Oryza_sativa*.IRGSP-1.0.pep.all.fa.gz) as queries against the *A. thaliana* protein database obtained from Ensembl Plants (release 56, *Arabidopsis_thaliana*.TAIR10.pep.all.fa.gz). The BLASTP search was conducted with an *E*-value cutoff of 1e−10. For each *O. sativa* gene, the top hit in the *A. thaliana* protein database was considered the putative ortholog. The corresponding "Gene ID," "Gene name," and "Gene description" for the *A. thaliana* orthologs were obtained from the Ensembl Plants (release 56, *A. thaliana* genes (TAIR10)).

## 3. Results

### 3.1. Overview of this study

Initially, we present an overview of the entire study. Through comprehensive data analysis, this research aimed to achieve two primary objectives:

(1) Identification of genes suggested to contribute to stress tolerance phenotypes in *O. sativa*.

(2) Identification of genes implicated by the data to be involved in common stress response mechanisms across plant species, specifically in *O. sativa* and *A. thaliana*.

This study consisted of six steps as shown in Figure 1.

### 3.2. Retrieval and curation of RNA-Seq data from public databases

RNA-Seq data were retrieved from the National Center for Biotechnology Information Gene Expression Omnibus (NCBI GEO) and European Bioinformatics Institute BioStudies (EBI BioStudies) public databases. Compared with microarray data, RNA-Seq data were primarily derived from the Illumina platform, making them more suitable for comparative analyses of studies from different research groups. Therefore, only RNA-Seq data were used in this study, while microarray data were excluded. In this study, based on the description in the papers and metadata of databases, the phenotype of cultivars reported to present strong resistance was defined as "resistant" and the cultivars reported to present relatively weak resistance were defined as "susceptible." A total of 202 samples were collected from the 11 projects, resulting in 105 paired datasets of stress-treated and untreated control samples. Because of manual curation, the collected data were divided into four categories based on the stress type and phenotype: (1) salt-resistant, (2) salt-susceptible, (3) drought-resistant, and (4) drought-susceptible. The number of paired datasets for each category was as follows: 24 for salt-resistant, 36 for salt-susceptible, 31 for drought-resistant, and 14 for drought-susceptible. To systematically organize the data used in the analysis, a table including the stress type, phenotype, subspecies, research project ID, and experimental pair number was prepared for each rice variety (Supplementary Table S1). When the number of collected sample pairs was compared among the tissues, it was revealed that samples derived from "leaf" contained the most

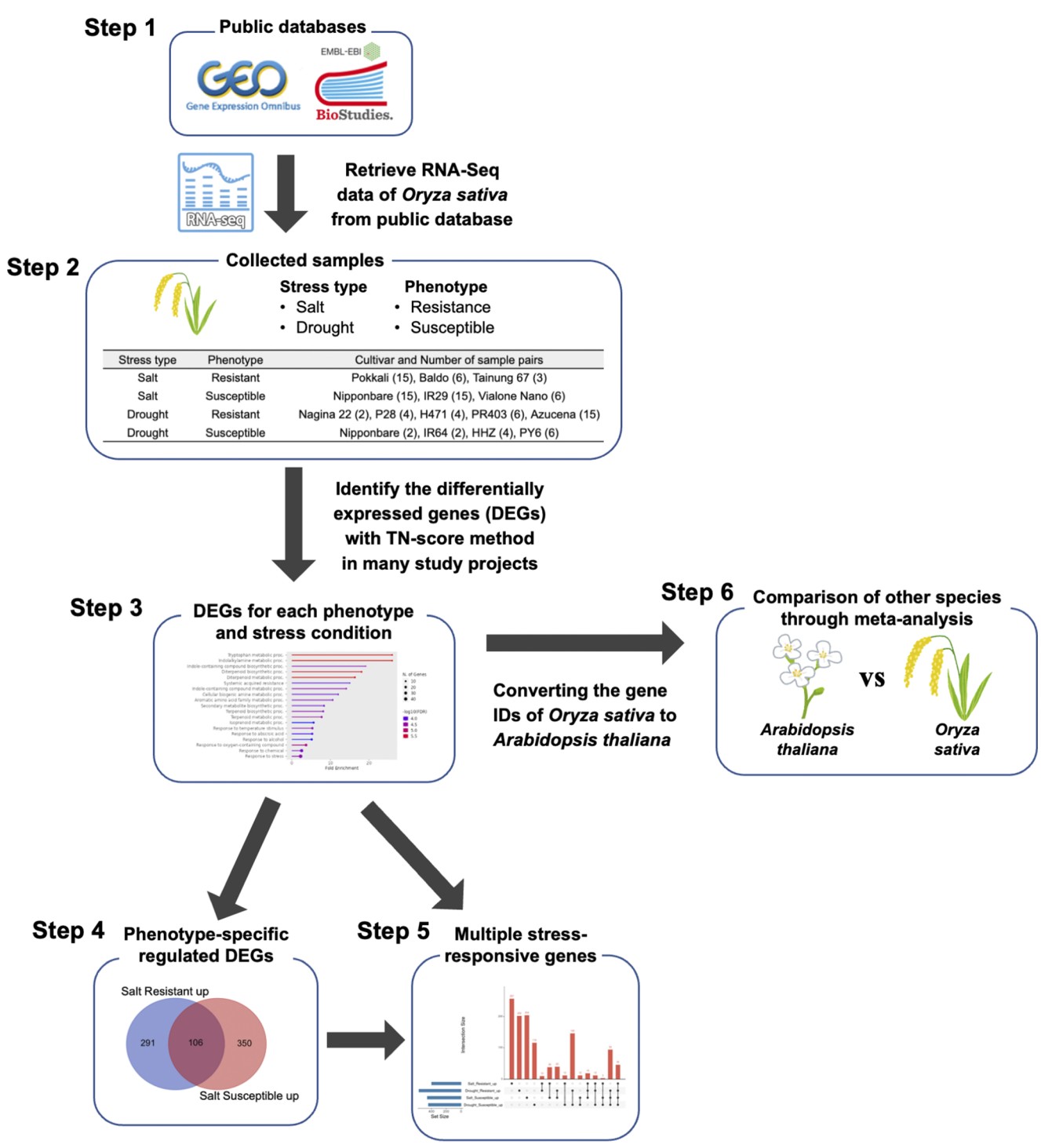

**Figure 1.** Overview of the six steps of this study. Step 1: RNA-Seq data for *Oryza sativa* were retrieved from public databases (National Center for Biotechnology Information Gene Expression Omnibus (NCBI GEO) and European Bioinformatics Institute BioStudies (EBI BioStudies). Step 2: Samples were categorized according to stress type (salt/drought) and phenotype (resistant/susceptible). Differentially expressed genes (DEGs) were identified using the TN-score method. Step 3: Enrichment analysis was performed on the DEGs for each phenotype and stress condition to evaluate the validity of the analysis. Step 4: Genes involved in the phenotypic differences between *O. sativa* cultivars were selected under drought and salt stress conditions. Step 5: Common genes among different stress conditions were selected. Step 6: Genes involved in stress response mechanisms conserved across different plant species were selected.

data (Supplementary Figure S1). Details on all other metadata are presented in Supplementary Table S2.

### 3.3. Analysis of collected data and selection of differentially expressed genes in O. sativa

The collected samples were subjected to quality control and expression quantification using Salmon(Patro et al., 2017), which calculated the transcripts per million (TPM) values for each gene in each sample (Supplementary Table S3). Stress-treated and non-treated samples were paired for each gene, and the expression ratios (TN ratios) and TN scores were calculated. The TN ratio was calculated as follows:

$$TN\ ratio = (stress\text{-}treated\ TPM + 1) / (non\text{-}treated\ TPM + 1).$$

If the TN ratio was higher than the threshold, the gene was considered upregulated; whereas if it was lower than the reciprocal of the threshold, it was considered downregulated. Otherwise, it was considered unchanged. To classify the upregulated and downregulated genes, we evaluated 1.5-fold, 2-fold, 5-fold, and 10-fold thresholds and finally chose the 2-fold threshold. Therefore, genes with a TN ratio higher than 2 were classified as upregulated, while genes with a TN ratio lower than 0.5 were classified as downregulated. The TN score of each gene was determined by subtracting the number of downregulated experiments from the number of upregulated experiments to assess changes in gene expression under stress conditions across the entire experiment.

The collected samples were classified into the four aforementioned categories based on the phenotype and type of stress treatment, and the TN scores for each category were calculated. After considering multiple expression ratio thresholds, we adopted a 2-fold threshold (TN2). This threshold was slightly lower to provide a comprehensive analysis. More severe scores for the 5-fold (TN5) and 10-fold (TN10) thresholds were also calculated and are listed in the Supplementary Table. The TN ratios and TN scores for all the genes are available online (Supplementary Tables S4 and S5).

Additionally, the distribution of TN scores for the genes was visualized using scatter plots (Supplementary Figure S2). Focusing on points where scores changed significantly on the scatter plot, the top and bottom genes based on TN2 scores were selected as differentially expressed genes (DEGs). This accounted for the top or bottom 1% of all reference genes. Abrupt score changes indicated that the expression of these genes was remarkably variable in the dataset used in this study. The number of identified genes and range of TN2 scores are summarized in Table 1. Lists of the

upregulated and downregulated genes are available online (Supplementary Tables S6 and S7, respectively). For the DEGs identified in rice, the corresponding orthologs in *Arabidopsis thaliana* were determined using sequence similarity searches with BLASTP. The list of these genes is available online (Supplementary Table S8).

### 3.4. Comparison of DEGs Identified by TN2 and DESeq2

Similarly, we identified differentially expressed genes (DEGs) using DESeq2 on salt-resistant rice samples comprising 24 salt-treated and 24 untreated specimens. In our DESeq2 analysis, genes were considered differentially expressed if their expression changed by at least a twofold ratio ($\log^2$ fold change $\geq 1$) and met specific false discovery rate (FDR) thresholds. We tested four FDR values – 0.05, 0.01, 0.005, and 0.001 – and at FDR < 0.05, DESeq2 identified 82 upregulated genes. To evaluate the concordance between DESeq2 and the TN-score method, we compared the DEGs from DESeq2 with the 397 genes selected via the TN-score approach (Supplementary Table S9). Notably, 54 of the 82 upregulated genes (approximately 65.8%) identified by DESeq2 overlapped with the TN-score selection. This overlap substantiates the efficacy of the TN-score methodology by demonstrating its ability to capture a substantial proportion of genes exhibiting statistically significant expression changes while also identifying many additional genes not detected by DESeq2.

To address the biological significance of genes identified by the TN-score method but missed by DESeq2, Gene Ontology (GO) enrichment analysis was performed on DEG sets. These included those commonly identified by both methods and those identified by either method alone (Supplementary Table S10, Supplementary Figure S3). The results revealed that the gene set uniquely identified by the TN-score method (343 genes) showed significant enrichment in multiple GO terms related to important biological functions, such as the stress response (e.g., Cold acclimation, Response to cold, Response to water deprivation and Response to abscisic acid), compared to the gene set uniquely identified by DESeq2 (28 genes). This suggests that the TN-score method captures a biologically meaningful set of genes overlooked by DESeq2. Notably, the TN-score-specific DEGs included four Dehydrin proteins, *RAB16A* (*Os11g0454300*), *RAB16B* (*Os11g0454200*), *RAB16C* (*Os11g0454000*), and *RAB16D* (*Os11g0453900*), which were classified under all the aforementioned stress-related GO terms. Although the precise molecular functions of Dehydrins are still not fully understood, these Group II Late Embryogenesis Abundant (LEA) proteins are characterized by high hydrophilicity (Liu et al., 2017; Sun et al., 2021; Szlachtowska

**Table 1.** Distribution of DEG numbers and TN2 scores in *Oryza sativa* under different stress conditions and phenotypes based on the meta-analysis

| Treatment type | Phenotype | Regulation | All pairs | TN score (S) | Number of *O. sativa* genes | Number of ortholog *Arabidopsis* genes |
|---|---|---|---|---|---|---|
| Salt | Resistant | Upregulated | 24 | $6 \leq S \leq 15$ | 397 | 307 |
| Drought | Resistant | Upregulated | 31 | $13 \leq S \leq 26$ | 569 | 421 |
| Salt | Susceptible | Upregulated | 36 | $10 \leq S \leq 22$ | 456 | 312 |
| Drought | Susceptible | Upregulated | 14 | $9 \leq S \leq 13$ | 441 | 264 |
| Salt | Resistant | Downregulated | 24 | $-17 \leq S \leq -5$ | 473 | 379 |
| Drought | Resistant | Downregulated | 31 | $-22 \leq S \leq -11$ | 415 | 314 |
| Salt | Susceptible | Downregulated | 36 | $-16 \leq S \leq -6$ | 489 | 302 |
| Drought | Susceptible | Downregulated | 14 | $-14 \leq S \leq -7$ | 556 | 282 |

& Rurek, 2023). Their expression is induced in response to abiotic stresses such as salt, drought and low temperature, and they are believed to contribute to plant stress tolerance through mechanisms including the stabilization of biological membranes, protection of enzyme activity, or binding to metal ions. Plants accumulate highly hydrophilic dehydrins, which enable them to retain water and reduce cellular dehydration and damage caused by osmotic pressure, especially under conditions that disrupt intracellular water homeostasis, such as high salinity. Furthermore, the TN-score-specific DEGs also contained other genes belonging to families involved in salt and drought stress responses, such as *OsPP2C50* (*Os05g0537400*), *OsABA45* (*Os12g0478200*), *OsPSY* (*Os09g0555500*), *OsPP2C30* (*Os03g0268600*) and *LEA17* (*Os03g0322900*). These uniquely identified genes provide the evidence for the efficacy of the TN-score method in identifying biologically important stress-responsive genes that have not been detected by DESeq2 approaches.

Overall, the complementary use of the TN-score method alongside conventional statistical approaches offers an effective strategy for comprehensive gene expression analysis in meta-studies. The read count data used for the DESeq2 analysis and the corresponding DEGs are listed in Supplementary Table S9.

### 3.5. Enrichment analysis to evaluate the characteristics of DEGs

To evaluate the characteristics of the identified DEGs, we performed enrichment analyses of individual DEG groups using ShinyGO. The results are shown in Supplementary Figure S4 and Supplementary Table S11. The top two GO terms for each DEG are listed in Supplementary Table S12.

Enrichment analysis revealed distinct patterns of gene regulation under different stress conditions and phenotypes. These results suggest that diverse stress response mechanisms may exist depending on the type of stress and phenotypic traits.

Furthermore, we focused on DEGs that showed similar expression patterns (upregulation or downregulation) across different phenotypes and stress conditions and considered them as a single group. We performed enrichment analyses for two types of combined DEGs. The first group consisted of all genes whose expression was upregulated under at least one condition (Supplementary Figure S5a), and the second group consisted of all genes whose expression was downregulated under at least one condition (Supplementary Figure S5b).

For upregulated DEGs (Supplementary Figure S5a), the most significantly enriched terms were "diterpenoid metabolic processes" and "water deprivation." Other enriched terms included responses to various abiotic stressors, such as heat and salt stress, and hormone responses, especially abscisic acid (ABA). These results support the validity of the selection of DEGs responsive to salt and drought stress by the meta-analysis.

For the downregulated genes (Supplementary Figure S5b), the analysis revealed a different set of enriched terms. The most significantly enriched terms were related to nitrogen metabolism, including "nitrate metabolic processes," "nitrate assimilation," and "reactive nitrogen species metabolic processes." Interestingly, several terms involved in oxidative stress response and detoxification were also significantly enriched, such as "hydrogen peroxide catabolic process" and "reactive oxygen species metabolic process." Additionally, photosynthesis-related processes and transmembrane transport were among the enriched pathways for downregulated genes. Detailed results of the enrichment analysis

for individual DEGs and all combined DEGs are provided in Supplementary Table S9.

### 3.6. Selection of genes involved in phenotypic differences between O. sativa cultivars under drought and salt stress conditions

We compared the DEGs between the stress-resistant and stress-susceptible phenotypes for both salt and drought stress to identify genes specific to each condition (Supplementary Figure S6). The numbers of common and unique genes in each group are summarized in Supplementary Table S13.

We conducted an enrichment analysis of these individual gene sets, and the results are shown in Supplementary Figure S6. However, as no hits were found in gene set (j), only those for the "All available gene sets" setting were used. The top GO terms for each DEG are summarized in Supplementary Table S14.

Interestingly, the enrichment analysis showed that the most significantly enriched term for both the upregulated genes in the salt susceptible and salt resistant groups, as well as the drought susceptible and drought resistant groups, was "GO:0009631 Cold acclimation." This suggests that the importance of stress mechanisms of cold acclimation may be similar to that of both salinity and drought stress, regardless of the *O. sativa* phenotype.

In addition, in Supplementary Figure S6a, the most significantly enriched term in the enrichment analysis of the upregulated combined DEGs was "diterpenoid metabolic processes." In contrast, the most significantly enriched term in the genes specifically downregulated in the drought-resistant phenotype was "diterpenoid biosynthesis (KEGG: osa00904)." This suggests that the pathways involving diterpenoids may regulate stress responses by upregulating or downregulating the expression of related genes. All gene lists, Venn diagrams visualizing overlapping genes, and enrichment results are shown in Supplementary Tables (Supplementary Tables S15 and S16).

### 3.7. Selection of genes common among different stress conditions

The overlap of genes that were commonly upregulated or downregulated across the four categories based on the two stress conditions and two phenotypes each was visualized using UpSet plots (Figure 2).

All overlapping genes are listed in Supplementary Table 17. The number of genes whose expression specifically increased or decreased in each phenotype under both stress conditions is summarized in Supplementary Table S18.

Among these genes, those whose expression was upregulated in the resistant and susceptible varieties are summarized in Table 2.

In total, 10 genes were consistently upregulated in the resistant phenotype under both stress conditions. These included genes involved in various cellular processes, such as transcription regulation (*Oshox25* and *PHR3*), secondary metabolism (*OsOSC4* and *OsPSY*), and peroxidase (*prx130*).

In contrast, 12 genes were consistently upregulated in the susceptible phenotype under both salt and drought stress conditions. These genes have different functional profiles, including a bHLH transcription factor (*OsbHLH035*), glycine-rich cell wall structural protein (*GRP*), hypoxia-induced gene (*HIGD2*), and enzymes involved in various metabolic processes (*RRJ1* and *C10923*).

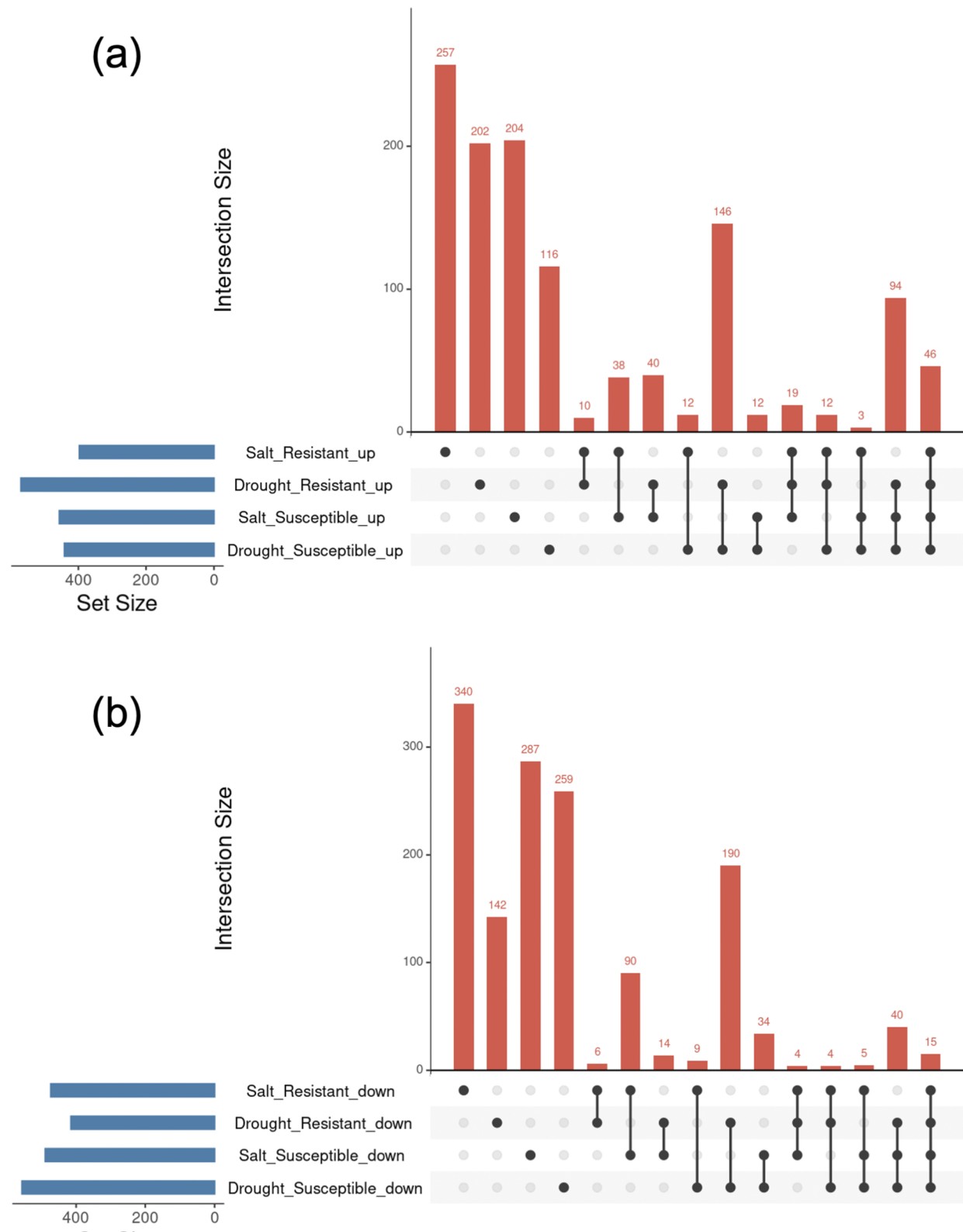

**Figure 2.** UpSet plots showing the overlap and specificity of DEGs in *Oryza sativa* under various stress conditions and phenotypes. UpSet plots illustrating the overlap of commonly or specifically regulated genes across the four categories combining salt and drought stress conditions with resistant and susceptible phenotypes. (a) Upregulated genes; (b) downregulated genes.

**Table 2.** List of *Oryza sativa* genes that were specifically upregulated in the resistant or susceptible phenotype under both salt and drought stress conditions

| Overlap | Total number of genes | Os-Gene stable ID | Os-Gene name | Os-Gene description |
|---|---|---|---|---|
| Drought_Resistant_up Salt_Resistant_up | 10 | Os12g0594000 | OsWD40–198 | WD40 repeat-like domain containing protein |
| | | Os09g0379600 | Oshox25 | Similar to homeobox-leucine zipper protein HOX25 |
| | | Os03g0709300 | OsUCL9 | Similar to chemocyanin precursor (basic blue protein) (plantacyanin) |
| | | Os02g0140400 | OsOSC4 | Similar to beta-amyrin synthase |
| | | Os02g0139000 | PHR3 | Transcription factor, regulation of Pi signaling and homeostasis, tolerance to low-Pi stress |
| | | Os08g0290700 | - | Winged helix repressor DNA-binding domain containing protein |
| | | Os11g0112400 | prx130 | Peroxidase (EC 1.11.1.7) |
| | | Os01g0767600 | - | Conserved hypothetical protein |
| | | Os09g0555500 | OsPSY3 | Similar to chloroplast phytoene synthase 3 |
| | | Os01g0155800 | - | Conserved hypothetical protein |
| Drought_Susceptible_up Salt_Susceptible_up | 12 | Os01g0159800 | OsbHLH035 | Similar to DNA binding protein |
| | | Os05g0119100 | - | Hypothetical protein |
| | | Os02g0209400 | - | Conserved hypothetical protein |
| | | Os10g0450900 | GRP | Similar to glycine-rich cell wall structural protein 2 precursor |
| | | Os10g0517500 | RRJ1 | Cys/Met metabolism, pyridoxal phosphate-dependent enzyme domain containing protein |
| | | Os12g0510750 | - | Conserved hypothetical protein |
| | | Os09g0278000 | - | Hypothetical conserved gene |
| | | Os07g0673900 | HIGD2 | Hypoxia induced protein, early stage of hypoxia signaling |
| | | Os11g0533400 | - | Conserved hypothetical protein |
| | | Os11g0700900 | C10923 | Glycoside hydrolase, subgroup, catalytic core domain containing protein |
| | | Os03g0305100 | - | Similar to AMP-binding protein |
| | | Os03g0184550 | - | Similar to dihydroflavonol-4-reductase |

### 3.8. Selection of genes involved in stress response mechanisms conserved across different plant species

In a previous study, we performed a meta-analysis of *A. thaliana* under ABA, salt, and dehydration conditions and identified genes that were upregulated or downregulated under each condition (Shintani et al., 2024). Here, we used *A. thaliana* orthologous genes corresponding to the *O. sativa* genes identified in this study to determine whether any genes shared commonality. The overlap in expression of commonly regulated genes was visualized using UpSet plots (Supplementary Figure S7). Satisfying the above criteria, 11 genes were upregulated (At_ABA_up, At_Dehydration_up, At_Salt_up, Os_Drought_Resistant_up, Os_Drought_Susceptible_up, Os_Salt_Resistant_up, and Os_Salt_Susceptible_up) and 1 gene was downregulated (At_ABA_down, At_Dehydration_down, At_Salt_down, Os_Drought_Resistant_down, Os_Drought_Susceptible_down, Os_Salt_Resistant_down, and Os_Salt_Susceptible_down) under salt and drought conditions (Table 3). These genes may be involved in the stress response mechanisms that are common across different plant species.

For the *O. sativa* genes in the 'resistant' phenotype category used in this study, six genes were upregulated (At_ABA_up, At_Dehydration_up, At_Salt_up, Os_Drought_Resistant_up, and Os_Salt_Resistant_up) and one gene was downregulated (At_ABA_down, At_Dehydration_down, At_Salt_down, Os_

Drought_Resistant_down, and Os_Salt_Resistant_down) was downregulated under salt and drought conditions (Table 3). These genes may be involved in stress responses that are common across plant species, particularly in mechanisms important for stress tolerance.

The list of *O. sativa* gene IDs identified in this study was converted to *A. thaliana* IDs and compared with previous studies. All gene lists are available online (Supplementary Table S19).

## 4. Discussion

This study aimed to identify novel genes and potential pathways involved in drought and salt stress responses in *O. sativa* using a meta-analysis of publicly available RNA-Seq data. By integrating datasets from multiple studies covering different *O. sativa* cultivars and stress conditions, we aimed to reveal robust and consistent gene expression patterns associated with stress resistance or susceptibility. Analysis of the tissue types of the collected samples revealed that while salt and drought stress are expected to have significant effects on the roots, the majority of the affected samples in this study consisted of "leaf" and "shoot" tissues, with a limited number of "root-"specific samples. Although this study focused on phenotypes and stress types rather than tissue specificity, tissue-specific analyses will likely become possible in the future as the amount of publicly available RNA-Seq data increases.

**Table 3.** Comparison of ABA, salt, and drought stress-responsive genes conserved in rice and *Arabidopsis*

| Overlap | Total number of genes | At-gene stable ID | At-gene Name | At-description |
|---|---|---|---|---|
| At_ABA_up At_Dehydration_up At_Salt_up Os_Drought_Resistant_up Os_Drought_Susceptible_up Os_Salt_Resistant_up Os_Salt_Susceptible_up | 11 | AT1G52690 | LEA7 | Late embryogenesis abundant protein (LEA) family protein |
| | | AT3G03341 | - | Cold-regulated protein |
| | | AT1G07430 | HAI2 | highly ABA-induced PP2C protein 2 |
| | | AT2G35300 | LEA18 | Late embryogenesis abundant protein, group 1 protein |
| | | AT3G48510 | - | AtIII18x5-like protein |
| | | AT2G39050 | EULS3 | hydroxyproline-rich glycoprotein family protein |
| | | AT3G24520 | HSFC1 | heat shock transcription factor C1 |
| | | AT2G46270 | GBF3 | G-box binding factor 3 |
| | | AT5G66400 | RAB18 | Dehydrin family protein |
| | | AT2G47770 | TSPO | TSPO(outer membrane tryptophan-rich sensory protein)-like protein |
| | | AT2G46680 | HB-7 | homeobox 7 |
| At_ABA_down At_Dehydration_down At_Salt_down Os_Drought_Resistant_down Os_Drought_Susceptible_down Os_Salt_Resistant_down Os_Salt_Susceptible_down | 1 | AT5G05440 | PYL5 | Polyketide cyclase/dehydrase and lipid transport superfamily protein |
| At_ABA_up At_Dehydration_up At_Salt_up Os_Drought_Resistant_up Os_Salt_Resistant_up | 6 | AT4G19230 | CYP707A1 | cytochrome P450, family 707, subfamily A, polypeptide 1 |
| | | AT1G60190 | PUB19 | ARM repeat superfamily protein |
| | | AT1G45249 | ABF2 | Abscisic acid responsive elements-binding factor 2 |
| | | AT3G63060 | EDL3 | EID1-like 3 |
| | | AT1G29640 | - | senescence regulator (Protein of unknown function, DUF584) |
| | | AT1G48000 | MYB112 | myb domain protein 112 |
| At_ABA_down At_Dehydration_down At_Salt_down Os_Drought_Resistant_down Os_Salt_Resistant_down | 1 | AT1G80050 | APT2 | Adenine phosphoribosyl transferase 2 |

This study focused on several *O. sativa* cultivars that exhibited different phenotypic responses to salt and drought stress. The phenotypic classifications of "resistant" and "susceptible" used in this study are based solely on existing research reports and may not fully capture the nuanced stress tolerance levels of each cultivar. In addition, differences in stress treatment methods among the studies should be considered.

To minimize the impact of these differences and provide a robust comparative analysis of multiple datasets, we adopted the following approach. First, we directly compared the gene expression ratios between stressed and control samples in the same study. Subsequently, we calculated the TN score, which facilitated integration and comparative analysis of the data across multiple studies.

Based on the TN score, DEGs were identified for each dataset and categorized by stress type and phenotype. Enrichment analysis of each DEG set revealed the distinct and characteristic terms associated with each DEG group. Furthermore, we performed an enrichment analysis of genes that were upregulated or downregulated, irrespective of the phenotypic classification or stress type. The most significantly enriched term for upregulated genes was "diterpenoid metabolic process," with 13 genes included in this category (Supplementary Figure S5 (b)). Diterpenoid phytoalexins have been shown to be potentially involved in the resistance of cereal plants to environmental stresses, such as drought (Murphy & Zerbe, 2020). The genes are listed in Supplementary Table S20.

These genes included those whose functions in stress mechanisms have been elucidated in previous studies. *OsCPS4* encodes a key enzyme that contributes to the biosynthesis of labdane-related diterpenoid (LRD) phytoalexins in *O. sativa*. Mutant analysis showed that the loss of function of *OsCPS4* significantly increased sensitivity to drought stress, suggesting that LRD biosynthesis involving *OsCPS4* contributes to the regulation of stomatal closure, independent of the ABA pathway (Zhang et al., 2021). The genes with unknown functions listed in this study may include those that contribute to the stress response through their involvement in LRD metabolism. Interestingly, in addition to the upregulated DEGs, the term enriched in genes specifically downregulated in the drought stress susceptible samples was "diterpenoid biosynthesis (KEGG: osa00904)" (Supplementary Figure S6 (d), Supplementary Table S20). This contrasting expression pattern suggests a complex

regulatory mechanism in the diterpenoid pathway. In other words, the genes in this pathway may show diverse expression patterns depending on the type of stress, plant phenotype, and cultivar.

We also focused on a group of genes whose expression was upregulated in both the resistant and susceptible cultivars under the two stresses. Some of the genes commonly upregulated under salt and drought stress in the resistant phenotype included *OsPSY3* (*Os09g0555500*). *PSY* is a rate-limiting enzyme in carotenoid synthesis and represents a precursor required for ABA synthesis in various plants (Zhou et al., 2022). *OsPSY3* has been demonstrated in previous studies using reverse transcription quantitative PCR (RT-qPCR) experiments to show upregulation of gene expression after dehydration, salt, and ABA treatment (Du et al., 2015; Welsch et al., 2008). Therefore, the selection results were consistent with those of existing studies. The identification of this gene as one of the stress-responsive genes in the previous studies provides evidence supporting the results of the meta-analysis. Other upregulated genes included transcription factor *OsPHR3* (*Os02g0139000*). This gene is also upregulated in response to dehydration stress and confirmed by RT-qPCR in the previous study (Abdirad et al., 2022). The expression of this gene is also induced by inorganic phosphate (Pi) deficiency and plays a role in improving tolerance to Pi deficiency as well as regulating nitrogen (N) homeostasis (Guo et al., 2015; Ruan et al., 2017; Sun et al., 2018). Deficiency of Pi or N induces various morphophysiological adaptive responses and gene expression regulation in plants (Bechtaoui et al., 2021; Khan et al., 2023; Sakuraba, 2022; Xing et al., 2023). Although several studies have suggested a relationship between Pi and N deficiency and the ABA pathway and its biosynthesis, many aspects remain unclear and further research is needed (Fang Zhu et al., 2018; Hsieh et al., 2018; Jaskolowski & Poirier, 2024; Zakari et al., 2020; Yu Zhang et al., 2022). The functions of these genes, specifically upregulated in resistant cultivars, suggest that they contribute to tolerance mechanisms. For example, the consistent upregulation of *OsPSY3*, a key enzyme in the carotenoid biosynthesis pathway that leads to the production of ABA, may enable resistant cultivars to synthesize ABA more quickly in response to stress. This could result in more efficient stomatal closure, reducing water loss and enhancing osmotic adjustment, thereby conferring drought and salt resistance. Similarly, the upregulation of *OsPHR3*, which regulates Pi homeostasis, could be crucial for maintaining the balance of nutrients under stress. As abiotic stresses often disrupt nutrient uptake and homeostasis, sustaining phosphate and potentially related nitrogen signaling pathways via *OsPHR3* could enable resistant plants to manage resources and maintain growth more effectively than susceptible ones.

In contrast, genes specifically upregulated in susceptible cultivars included *OsbHLH035* (*Os01g0159800*) and *RRJ1* (*Os10g0517500*). A previous meta-analysis study using transcriptome data suggested that genes such as *OsbHLH035* and *RRJ1* were upregulated by multiple heavy metal stresses, including mercury (Hg) and lead (Pb), and were subsequently validated by RT-qPCR (Fan et al., 2021). Drought-induced reductions in soil moisture can increase the concentration of heavy metals in the soil, which can have a synergistic negative effect on plants (Islam & Sandhi, 2022). Therefore, focusing on genes such as *OsbHLH035* and *RRJ1* is likely to provide important clues for understanding the combined effects of heavy metal stress and drought stress and for developing resistant varieties. These genes specifically upregulated in susceptible cultivars might not signify an optimal adaptive response to salt or drought. It is plausible that their induction, potentially reflecting pathways shared with heavy metal stress,

leads to suboptimal resource allocation or triggers inappropriate signaling cascades, thereby contributing to susceptibility. An alternative hypothesis is that the higher expression is merely a consequence of greater overall stress damage experienced by susceptible cultivars, resulting in a more general, less specific stress response activation that includes these genes. However, it should be noted that experimental verification of the expression of these genes under salt or drought treatment has not yet been conducted.

Additionally, a comparison with previous *Arabidopsis* meta-analysis data revealed several genes whose expression was altered in *O. sativa* and *A. thaliana* under both salt and drought stress conditions (Table 3). To focus on genes involved in mechanisms commonly conserved between plant species regardless of the stress phenotype, we focused on genes shared by seven DEGs, with three in *A. thaliana* and four in *O. sativa*. For example, *HSFC1* (*AT3G24520*) represented a gene commonly upregulated in both species, regardless of phenotype.

Previous studies on *A. thaliana* have suggested that the expression of *HSFC1* is involved in cold responses via a pathway independent of the C-repeat binding factor (CBF), a well-known transcription factor that controls cold acclimation in plants (Jia et al., 2016; Zhao et al., 2016).

The ortholog of *Arabidopsis HSFC1* gene in *O. sativa* was *HSfC2b* (*Os06g0553100*). However, the functions and roles of these genes under salinity and drought conditions remain unclear. Heat shock factors (HSFs) that contribute to environmental stress tolerance occur in both *A. thaliana* and *O. sativa* (Wenjing et al., 2020; Yan Zhang et al., 2022). Given the consistent upregulation of *HSFC1* and its ortholog *HSfC2b* across different stress conditions and plant species, these genes are promising candidates for further investigation.

In the cross-species comparison, the relatively limited overlap of common genes can be primarily attributed to the fact that we focused on genes shared by multiple DEG sets – specifically, three sets in *A. thaliana* and four sets in *O. sativa*, totaling seven sets. Furthermore, *O. sativa* and *A. thaliana* are evolutionarily highly divergent and possess distinct stress response pathways, making it natural that only a few genes exhibit similar expression changes under all conditions. Nevertheless, the genes commonly identified in both species are likely universally conserved and function as core regulatory factors in stress responses, underscoring their significant biological relevance.

Regarding the GO term "cold acclimation" upregulated in both resistant and susceptible groups, upon examining the list of genes annotated with this term, we observed that the majority were dehydrin genes. Dehydrins contribute to drought and salt stress tolerance by being induced not only by low temperatures but also by dehydration resulting from salt and drought stresses (Hanin et al., 2011). This suggests that the significant enrichment of "cold acclimation" in both resistant and susceptible groups largely reflects the activation of a common dehydration response mechanism rather than a cold-specific response.

This study had some limitations. First, because the identification of DEGs was based on the selection of TN scores rather than statistical tests, the results should be interpreted with caution. Second, the public RNA-Seq data used in this study is an integration of data obtained under various growth conditions and experimental protocols. As a result, there are variations in stress treatment methods between experiments, and we are constrained by the necessity to rely on published metadata information for phenotype classification. The classification of resistant and susceptible phenotypes was determined primarily based on metadata. Although this practical

method enabled the meta-analysis across heterogeneous datasets, it may not fully reflect the complete spectrum of stress tolerance mechanisms. These points need to be considered. Third, experimental validation using plant samples exposed to specific stress environments was not performed, indicating that further research is required to elucidate the functions of these genes. Fourth, we utilized the *Japonica* reference transcriptome (IRGSP-1.0). This decision was based on several factors, including that it provides more comprehensive and continuously updated gene annotations compared to the currently available Indica reference transcriptome. This decision was based on the fact that information about each gene is easier to access for *Japonica* compared to Indica. In particular, when discussing the functional interpretation of individual candidate genes identified in this meta-analysis, the information provided by Oryzabase (Kurata & Yamazaki, 2006) and RAP-DB (Sakai et al., 2013), which are major databases providing access to functional information and research papers on rice genes, is much more extensive for *Japonica* than for Indica. A key point here is that the TN ratio, the primary analytical metric in this study, is calculated within each experimental project as the ratio of gene expression levels (TPM) between paired stressed and control samples. This means that for datasets of Indica cultivars, the expression ratio is calculated between Indica stressed samples and Indica control samples. Therefore, we are not directly comparing absolute expression values between *Japonica* and Indica across studies or cultivars, but rather focusing on the relative expression changes during stress response within each experiment. Consequently, even if a mismatch between our *Japonica* reference sequence and Indica samples affects absolute TPM values, as long as both treated and control groups are similarly affected, the impact on their ratio, i.e., the TN ratio, is considered to be relatively small. Furthermore, the TN score method is a system that assigns higher scores to genes whose expression changes are consistently observed across a large number of datasets. In relation to the use of this *Japonica* reference, when comparing mapping to the Indica reference available on Ensembl Plants and the *Japonica* reference (IRGSP-1.0), 102 out of the 110 Indica samples used in this study indeed showed higher mapping rates to the *Japonica* reference (Supplementary Table S21). While this suggests a certain validity in selecting the *Japonica* reference, it is still possible that mapping efficiency could be reduced in some samples due to reference sequence mismatch. However, in the TN score method, even if mapping efficiency is low for some samples, if consistent expression changes are not detected in many other pairs, the specific variation in that sample is, in principle, unlikely to significantly impact the overall conclusions. In fact, for the drought-resistant category, when comparing the results from analyses excluding 2 pairs, 4 samples with low mapping rates (SRR11342745, SRR11342746, SRR11342749, SRR11342750) from its total of 31 paired datasets, versus using all samples for this category, over 95% of the DEGs were common (Supplementary Table S22a, Supplementary Figure S8a). Additionally, this study mainly used samples of leaves and shoots, with relatively few samples derived from roots. Therefore, due to the principle of TN score calculation, expression patterns specific to tissues with fewer samples, such as roots in this study, tend to have a relatively lower contribution to the overall score. We also compared the analysis results for the salt susceptible category when its root samples were excluded versus when all its tissues were used (Supplementary Table S22b, c, d)). For this salt susceptible category, when comparing the DEG results from the analysis excluding its 6 pairs of root samples (from its total of 36 paired datasets) with the analysis using all tissues for this category, approximately 70% of

the genes were common. Enrichment analysis of these common genes for the salt-susceptible category confirmed that similar terms were enriched as in the analysis using all tissues for this category (Supplementary Figure S8b, 8c, 8d)). However, the genes identified in this study are likely to reflect changes predominantly observed in leaves and shoots, which are the majority in our analyzed datasets, or changes common across tissue types. These limitations should be considered when interpreting our results and planning future studies.

## 5. Conclusion

Through a meta-analysis of publicly available RNA-Seq data, we obtained a list of potential target genes related to drought and salt stress responses in *O. sativa* and identified candidate genes that may influence stress-related phenotypes. These findings provide a valuable resource for future studies aimed at understanding the molecular mechanisms underlying stress resistance in rice and developing new strategies to improve crop productivity under adverse environmental conditions.

**Open peer review.** To view the open peer review materials for this article, please visit http://doi.org/10.1017/qpb.2025.10020.

## Acknowledgements

Computations were performed at the Hiroshima University Genome Editing Innovation Center.

**Competing interest.** The authors declare that this study was conducted in the absence of any commercial or financial relationships that could be construed as potential conflicts of interest.

**Data availability statement.** The data presented in this study are publicly available at Figshare (https://doi.org/10.6084/m9.figshare.25547083.v4).

**Author contributions.** Conceptualization, M.S. and H.B.; Methodology, M.S. and H.B.; Software, M.S. and H.B.; Validation, M.S. and H.B.; Formal Analysis, M.S. and H.B.; Investigation, M.S.; Resources, M.S. and H.B.; Data Curation, M.S.; Writing – Original Draft Preparation, M.S.; Writing – Review and Editing, M.S. and H.B.; Visualization, M.S.; Supervision, H.B.; Project Administration, H.B.; Funding Acquisition, H.B. All authors have read and agreed to the published version of the manuscript.

**Funding statement.** The authors declare that they received financial support for the research, authorship, and publication of this article. This work was supported by the Center of Innovation for Bio-Digital Transformation (BioDX), an open innovation platform for industry–academia cocreation (COI-NEXT), the Japan Science and Technology Agency (JST) (grant number JPMJPF2010).

**Supplementary materials.** The Supplementary Material for this article can be found online at https://doi.org/10.6084/m9.figshare.25547083.v4.

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
