## [Reviewer Report]

OVERVIEW

This manuscript presents a meta-analysis of public RNA-seq data to identify genes involved in drought and salt stress responses in rice varieties with different stress tolerance levels. The study successfully integrates multiple datasets to reveal consistently differentially expressed genes and explores conserved mechanisms through comparison with Arabidopsis.

STRENGTHS

Comprehensive integration of data across different experimental conditions

Identification of conserved stress response mechanisms between rice and Arabidopsis

Validation approach through cross-species comparison with previous findings

ISSUES REQUIRING REVISION

Title and Terminology

a) The term “phenotypes” in the title is imprecise and could be confused with other traits

b) Suggested revision: “Meta-analysis of Public RNA-sequencing Data of Drought and Salt Stresses in Resistant and Susceptible Oryza sativa Cultivars”

Methodological Concerns

Reference Genome Usage

a) Using japonica reference for both subspecies needs justification

b) Potential mapping biases due to sequence variations between japonica and indica

c) Gene presence/absence variations between subspecies should be addressed

Expression Analysis

a) TPM summarization method for multiple isoforms needs clarification, how is that performed?

b) TN ratio threshold (2/0.5) requires statistical justification

c) Distribution of TN values should be provided

d) Need for proper statistical testing for differential expression

Phenotype Classification

a) Criteria for resistant/susceptible phenotypic classification needs more detail

b) Potential inconsistency in phenotype definitions across studies should be addressed

Results Interpretation and Presentation

Gene Selection and Enrichment Analysis

a) Top GO term “cold acclimation” appears in both resistant/susceptible groups - needs biological interpretation

b) Small number of genes (11 up, 1 down) in cross-species comparison needs discussion on biological significance and biological meaning. Is this small overlap reasonable?

---

## [Editor Report]

Dear Dr. Hidemasa Bono, 

Your manuscript entitled "Meta-analysis of Public RNA-sequencing Data of Drought

and Salt Stresses in Different Phenotypes of Oryza sativa" has been peer-reviewed, and we are providing you with comments below. In summary, both the reviewer and the editor agree that the manuscript demonstrates a strong academic merit by conducting a meta-analysis to identify rice-specific and cross-species conserved DEGs in response to drought and salinity stresses. The scope of the meta-analysis is commendable, as it comprehensively covers a wide range of published datasets, and the results are further cross-referenced with the list of stress-related genes in A. thaliana published by the same research group. However, since this study relies solely on meta-analysis without functional validation, the methods presented require further clarification to strengthen the justification for the proposed list of genes for future functional studies. Please find the comments below that might be used to improve the manuscript.

---

## [Reviewer Report]

The authors have contextualized their findings on specific genes (OsPSY3, OsPHR3, OsbHLH035, RRJ1) by connecting them to previous experimental validation in the literature. This strengthens the credibility of their meta-analysis approach and demonstrates its ability to identify biologically relevant genes.

Their analysis of diterpenoid metabolism is particularly strong. They present a systematic view of how genes in this pathway display complex, sometimes contrasting expression patterns depending on stress conditions and plant phenotypes.

The cross-species comparison with Arabidopsis is well-executed and reveals conserved stress response mechanisms. Their focus on HSFC1/HSfC2b as a potential universal stress regulator across plant species represents a valuable insight for future research.

The authors have appropriately addressed limitations of their methodology, particularly regarding their reliance on the Japonica reference transcriptome and the TN score approach. Their discussion of the “cold acclimation” GO term enrichment shows good analytical reasoning to attribute this to the presence of dehydrin genes rather than a cold-specific response.

However, they could still strengthen several aspects:

The discussion of phenotype-specific genes could be expanded with more mechanistic hypotheses about how these genes might confer resistance or susceptibility

They could more explicitly address the biological significance of genes that were uniquely identified by their TN score method but missed by DESeq2

---

## [Reviewer Report]

This manuscript conducts an important analysis on drought and salt stresses using a meta-analysis of public RNA-seq data. However, the following points require reconsideration.

In the meta-analysis, it is crucial to minimize biases in the data that arise from technical factors, such as differences in experimental conditions. While the authors have taken the impact of instrumentation and other factors into account in their data selection criteria, they need to address two key points more thoroughly.

Firstly, as noted by Reviewer 1, the influence of the potential mapping biases due to sequence variations between japonica and indica should not be overlooked. For example, in Supplementary Table S20, some samples exhibit a mapping rate as low as 20.5%, suggesting that excluding these samples could significantly impact the results of the analysis. Although differences in gene presence/absence or sequence variations among subspecies are possible, it is essential to demonstrate that the observed differences in the candidate genes identified in this study are due to expression differences, rather than variations in genomic structure.

Secondly, while the authors have utilized the reference genome from the Ensemble Plants, more recent reference genomes are available in databases such as Gramene (https://oryza-ensembl.gramene.org/index.html). It would be beneficial to re-evaluate the analysis using these updated resources. Additionally, Supplementary TableS2 indicates that the authors have used data from various tissues, including leaf and root, in their analysis. Given that gene expression in response to stress can vary significantly between tissues like leaf and root, conducting tissue-specific analyses is important. A reanalysis that excludes data from root is recommended.

Addressing these issues is crucial to minimize the risk of false positives—apparent expression differences caused by batch effects when there is no true biological difference—and false negatives, where genuine expression differences may be obscured by batch effects and go undetected.

I hope these comments will be helpful.

---

## [Editor Report]

The revised manuscript by Shintani and Bono has made a substantial improvement in providing technical details of their analysis and contextualizing the findings in biological senses, providing the list of genes of interest for future research on stress responses in general. While there has been significant improvement by addressing all the comments from the reviewer 1, QPB managed to secure another set of comments from reviewer 2, who could independently evaluate the revised manuscript. Please find the comments that should be helpful in further improving the current manuscript. 

While the authors acknowledged the imitation of the work based on the skewed reference genome availability with high quality (japonica vs. indica), which is a general issue for meta-analysis work, we would like to invite the authors for another round of minor revision to address some technical points raised by reviewer2. There are comments on how to handle poor read mapping and tissue specificity, which I believe useful for the authors to perform the last step of sanity check to publicize this exciting finding. This proposed round of revision is expected to enhance scientific rigour. The comments for both reviewers can be hopefully utilised to revise and finalise this exciting work to be published at QPB. 

Thank you again for your consistent efforts to make the meta-analysis more addressable than ever with a highly rigorous manner.

---

## [Reviewer Report]

The authors have addressed all the reviewer comments thoroughly and appropriately. The revised manuscript reflects a clear and careful consideration of the feedback provided. All major and minor concerns raised during the review process have been adequately resolved. Therefore, I recommend that the manuscript be accepted for publication in its current form.